# Biphasic Injection for Masseter Muscle Reduction with Botulinum Toxin

**Fabrizio Chirico** [1,*], **Pierfrancesco Bove** [2], **Romolo Fragola** [3], **Angelo Cosenza** [4], **Nadia De Falco** [4], **Giorgio Lo Giudice** [5], **Giovanni Audino** [5] and **Giuseppe Mario Rauso** [6]

1   Private Practice, 80136 Naples, Italy
2   Private Practice, 80136 Naples, Italy; 20121 Milan, Italy; 00192 Rome, Italy; dr.pierfrancesco.bove@gmail.com
3   Maxillofacial Surgery Unit, Multidisciplinary Department of Medical-Surgical and Dental Specialities, University of Campania "Luigi Vanvitelli", 80138 Naples, Italy; romolofragola@gmail.com
4   General Surgery Unit, Department of Advanced Medical and Surgical Sciences, University of Campania "Luigi Vanvitelli", Piazza Miraglia, 80138 Naples, Italy; angelo.cosenza@unicampania.it (A.C.); nadia.defalco@outlook.it (N.D.F.)
5   Maxillofacial Surgery Unit, Department of Neurosciences, Reproductive and Odontostomatological Sciences, University Federico II, Via Pansini 5, 80100 Naples, Italy; Giorgio.logiudice@gmail.com (G.L.G.); audinodr.giovanni@gmail.com (G.A.)
6   Private Practice, 81055 Santa Maria Capua Vetere, Italy; centroapollonia@pec.it
*   Correspondence: fabriziochirico@hotmail.com; Tel.: +39-3283678855

**Abstract:** Masseter Muscle Hypertrophy (MMH) is a well-known clinical benign condition that is not gender-specific and it can be monolateral or bilateral. Botulinum Toxin type A (BoNTA) injection has been widely described for MMH treatment and non-surgical facial slimming. BoNTA masseter injections have high efficacy and safety profile, but the risks of side effects remain. Muscular bulging during mastication is a complication due to the superficial overcompensation of masseteric fibers in response to neurotoxic weakening of the deep masseter. We present a biphasic-injection technique for BoNTA administration, based following anatomical concept and developed in order to prevent paradoxical bulging. A total of 98 treatments from 2015 to 2020 were performed with this technique. No remarkable complications occurred in our study. No cases of loss of full smile, difficulty in mouth opening, dizziness, headache, neurapraxia, and xerostomia were reported. A case of asymmetric smiling was self-resolved within a week. No patient claimed transient muscle weakness as distressing. No cases of paradoxical bulging were observed. Extensive knowledge of muscular anatomy and appropriate injection technique are key factors in achieving the desired result and avoiding complications. We feel that sharing this tip could be helpful for all the physicians involved in MMH treatment with BoNTA.

**Keywords:** masseter muscle hypertrophy; botulinum toxin type A; biphasic injection technique; paradoxical bulging; muscular bulging; non-surgical facial slimming; lower face reshaping; neurotoxic weakening; complications; neurotoxic weakening

## 1. Introduction

The insight of facial attractiveness is influenced by quintessence and epitome of beauty. A wider lower third and a square jaw is aesthetically less desirable compared with an oval facial appearance. A slimmer lower third contributes to a heart-shaped face, which is becoming increasingly desired and considered more appealing, contextually emphasizing middle third prominence without any volume enhancement.

The contour of the lower third of the face is determined by the thickness of mandibular angle bone, soft tissues envelope, and masseteric muscle volume and projection. An unattractive squared-face appearance is commonly caused by a prominent mandibular angle or muscle enlargement as masseter hypertrophy, most common in people aged between 20 and 40 years [1].

Masseter muscle hypertrophy (MMH) is considered to be a well-known clinical, long recognized, and benign condition, in spite of the growing aesthetic concern manifested around it [2]. It is not gender-specific and it can be monolateral or bilateral. The etiology of masseter muscle hypertrophy has been attributed to a number of factors such as emotional stress, bruxism, clenching, temporomandibular joint disorders, masticatory muscles hyperactivity, masseteric hyperfunction and parafunction, and microtrauma as well as dietary habits [2–7]. Continuous gum, ethnic tough-food diets [2,8], and hard chewing can also lead to MMH [8,9]. Furthermore, malocclusion, masticatory muscle hyperactivity and mandibular retrognathia have been postulated as possible causes [1]. The origin of this condition is controversial because masticatory muscle hyperactivity or dysfunction and parafunction in stomatognathic system may not occur in all MMH cases [1–7]. Compensatory and stress hypertrophy could be assumed in many patients [4–6]. Modifications in proprioceptors have been debated [5,6]: cases of MMH in connection with neuroleptically-induced facial dystonia suggest that the masseteric hypertrophy could be attributed to a disturbance of the balance between neurotransmitter as acetylcholine and dopamine [7]. Emotional disturbances and psychological disorders impact the ability to keep muscular tone and proprioception, thus patients that suffer from those diseases are at a higher risk of developing MMH [5–7].

Various surgical and non-surgical treatments have been introduced for lower face reshaping. In the non-surgical therapeutic approach, effort has been made toward reducing the muscular hyperactivity by using occlusal splint or prescribing muscle relaxants [10]. Occlusal splints are easy to fabricate, widely used in the diagnosis and treatment of bruxism and temporomandibular disorders, and considered as the first choice for protecting teeth and prostheses from damages [11–14]. Moreover, Dalewski et al. describe that neither occlusal splint or modified nociceptive trigeminal inhibition splint device affected the masseteric muscle asymmetry or postural activity/maximum voluntary contraction ratio after treatment [11]. Nevertheless, occlusal devices can be effective in altering pressure pain threshold in patients with diagnosed bruxism [12].

Concerning surgical interventions, to achieve a slim and smooth lower facial contour, extraoral or intraoral approaches to perform ostectomy of the mandible along with surgical resection of the masseter have traditionally been recommended [4,15]. Gurney, in 1947 [4], tried to solve this condition performing masseteric resection through extra oral approach. In 1951, Converse [15] proposed an intraoral approach for the same purpose, associated with a resection of gonial angle. Moreover, radiofrequency was recommended to achieve volumetric reduction of the masseter muscle [16]. However, the drawbacks to these invasive interventions entail many risks such as facial nerve injury, infection, asymmetric resection, hematoma, uneven contour lines, persistent swelling, condyle fracture, postoperative pain, bleeding, scarring, and delayed healing time [4–16].

Although facial plastic surgery provides success in lower face reshaping, many patients prefer effective, minimally invasive alternative techniques [1–7,15–17]. As demonstrated by the American Society for Aesthetic Plastic Surgery statistical analysis, the demand for nonsurgical aesthetic procedures has increased during the last couple of decades. Even as the quantity of plastic surgery procedures doubled from 1995 to 2015, the amount of non-surgical cosmetic interventions increased by 12-fold [18–27]. This data demonstrates that among an increasing number of patients seeking for aesthetic improvements, non-surgical interventions are preferred over surgical procedures. As preferences for minimally invasive procedures are growing in cosmetic facial reshaping and according to the favorable safety profile along with no down-time, treatment with botulinum neurotoxin type A (BoNTA) has recently become a first-choice treatment for masseter hypertrophy instead of conventional surgery in lower face contouring.

Since it was first introduced in 1976 for strabismus treatment, BoNTA has been widely used to treat facial spasms in rehabilitation hubs [28]. In 1989, the U.S. Food and Drug Administration (FDA) approved the use of botulinum toxin for the treatment of facial spasmodic disorders [29,30]. While the toxin was being applied for the aforementioned

disorders, Carruthers suggested in 1987 that BoNTA could be used for cosmetic purposes to remove or ease facial wrinkles [30].

Aesthetic Plastic Surgery Statistics released in 2020 revealed that BoNTA are is still the first procedure performed in the top five nonsurgical approaches with an overall number of 2.643.366, 54.3% more than the year before [31,32]. These data undoubtedly confirm the increasing use of BoNTA, although are underestimated because they consider only the aesthetic use of BoNTA [33–35]. Currently, this drug is not used only to reduce wrinkles contractility and restore muscle tone but for the treatment of facial muscle hypertrophy, square-angled mandibula, migraine, headaches, hyperhidrosis, overactive bladder, blepharospasm, hemifacial spasm, oromandibular dystonia, cervical dystonia, focal limb dystonias, laryngeal dystonia, tics, essential tremor, bruxism management, and temporomandibular joint disorders [4–7,33–36].

Even if BoNTA injection cannot reduce the genesis of sleep bruxism, it can be considered as an elective management in reducing the intensity of masticatory muscles during bruxism [13]. Along with occlusal splints, BoNTA injection could result in protecting the masticatory system structures from the excessive force of sleep bruxism [13,14]. Shim et al. demonstrate the effect of BoNTA on jaw motor activity during sleep by a decrease of contractive intensity in the injected muscles [14].

Nowadays, BoNTA injection into masseter muscle is extensively used as an off-label non-invasive approach for MMH [37–39]. BoNTA acts selectively on the peripheral cholinergic motor nerve ending plate and inhibits acetylcholine release at neuromuscular junction [39]. Its effectiveness as an off-label drug in lower face reshaping has been proven by several studies [37–46]. The mechanism of action of BoNTA in masseter hypertrophy is different from wrinkle and hyperkinetic lines treatment. When injected at high concentrations, BoNTA may cause cell apoptosis, leading to atrophy of the masseter muscle. Moreover, it has been proved that BoNTA injections may prevent muscle fibers from regeneration and therefore muscle atrophy may be semi-permanent or even permanent [47,48]. Several studies have shown that jaw muscles are very different from other skeletal muscles and their regeneration after atrophy may be limited [47–49]. The combination of cell apoptosis in a muscle with limited regenerative capacities makes BoNTA an excellent treatment for masseter hypertrophy, without any relevant impact on mastication activities. Various studies demonstrate that botulinum toxin type A provides a long-term effect on the cosmetic reduction of masseter muscle volume to narrow lower face width [40,41]. As demonstrated by Rauso et al., the long-term stability of contouring of the lower face by injection of BoNTA could be achieved according to the occlusal therapy performed after the injection of BoNTA in patients showing widened lower face and malocclusion associated with masseter hypertrophy. The use of the occlusal device was performed to relieve muscle and to give a new guide for masticatory muscle: the new muscular tone is postulated to reduce the incidence of relapse of masseter hypertrophy thus to achieve long-lasting results [45]. Moreover, BoNTA acts as a muscular tension reliever: the retrograde transport of BoNTA and transcytosis block neurotransmitters diffusion across the peripheral nerve. BoNTA acts on nerve endings of the targeted muscle, thus an accurate and extensive anatomical knowledge of nerve endings is essential for achieving a maximum outcome with the minimum toxin concentration [44,45].

In 1994, Moore and Wood [42] first reported a non-surgical approach for the functional treatment of MMH, performed with BoNTA injections. Later, Rijsdijk et al. began to use botulinum toxin type A for aesthetic volumetric reduction of MMH [43]. Von Lindern et al., in 2001, confirmed the efficacy of BoNTA for MMH treatment as alternative treatment to surgery. Compared to invasive surgical techniques, BoNTA injection shows an excellent profile in terms of efficacy, safety, and no recovery time [1].

From 1994 to now, BoNTA injection has been widely described for MMH treatment; moreover, nowadays, non-surgical facial slimming is often performed with multiple procedures such as BoNTA injection into the masseter muscle [44,45]. Toxin masseter injections have a high efficacy and safety profile but the risks of a variety of side effects or complica-

tions such as bruising, headaches, smile limitation, and paradoxical bulging remain [46,50]. Thus, surgeons should be able to recognize those complications, their causes, and their prevention methods. Several anatomical studies have attempted to determine the most effective BoNTA injection points with the aim to achieve optimal results and minimize complications. Therefore, intramuscular nerve distribution, motor nerve entry point of masseteric nerve and the relationship between marginalis mandibular nerve and parotid gland were investigated [44,51–55]. Lee et al. reported the occurrence of paradoxical masseteric bulging after BoNTA injection [56].

Muscular bulging during mastication (also known as paradoxical bulging) occurs within a week after injection and is due to a superficial overcompensation of masseter muscle fibers in response to the neurotoxic weakening of the deep masseter [55]. Paradoxical bulging could be distressing for the patient, although it can be resolved, after 1–2 weeks from the previous injection, injecting some more units on the masseter. Even though this is a rare sequela, it is mandatory to focus on an injective approach based on anatomical clarification and performed to avoid this complication. We present a "biphasic" injection technique for BoNTA administration, developed in order to prevent paradoxical bulging.

## 2. Materials and Methods

From 2006 to 2014, with a total of 43 treatments performed on 21 patients, the senior author (P.B.) managed MMH with injection on 4 sites as already published in 2008 [44]. In 2014, for the first time, the senior author had a patient who experienced post toxin paradoxical bulging (Figure 1, Supplementary Materials: Video S1).

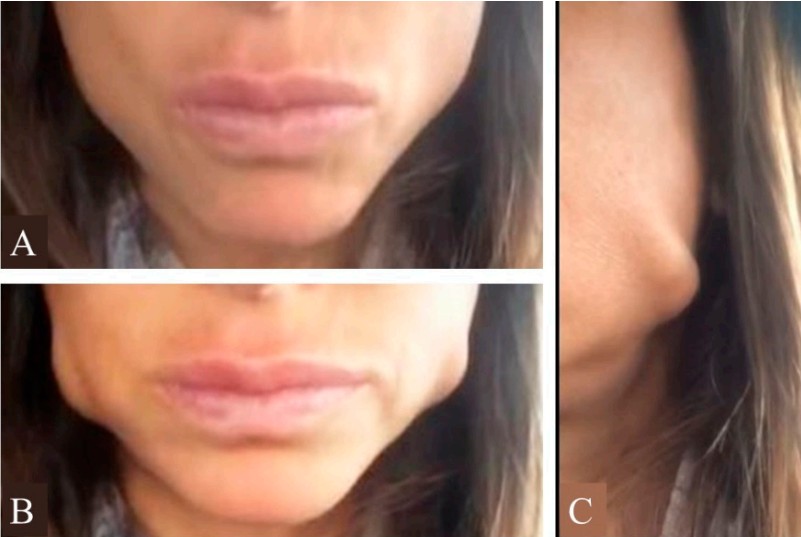

**Figure 1.** Morphologic aspect of bilateral masseter bulging on resting (**A**) and clenching (**B**) of the same patient showed in Video S1. Detail of masseteric belly affected by paradoxical bulging (**C**).

Since that moment, the author modified the injection technique in order to avoid this kind of complication. From the end of 2014, we started to deliver BoNTA into masseter muscle with a "biphasic" injection technique in order to avoid masseter paradoxical bulging. Abobotulinum toxin A (DYSPORT; Ipsen Ltd., Slough, UK), incobotulinum toxin A (BOCOUTURE; Merz Pharmaceuticals, Frankfurt, Germany) or onabotulinum toxin A (BOTOX; Allergan, an AbbVie company, North Chicago, IL, USA) have been used with no preferences: 0.8 mL of 0.9% saline were used for the reconstitution of 125 speywood units of Abobotulinum or 50 units of Inco- or Ona-botulinum.

Injections were performed using a 1-mL syringe with a 29 G 4 cm-long needle (Rays Health & Safety s.p.a., Osimo, AN, Italy). Because of variations of masseteric size and shape, we relied on palpation to delineate the anterior and posterior borders of the muscle, by asking the patient to bite down. The zygomatic bone and mandibular border were

marked. A zone of 1-cm intervals was drawn on the area overlying the low and lateral side of the jaw and a grid of 2-cm up from zygomatic arch to identify the area to be treated. Injections were performed on the crossed points, thus four site injections per side were given. According to the biphasic injection technique, 0.1 mL were released into the deep belly and 0.1 mL on the superficial belly of the masseter muscle at the same site injection. (Figures 2 and 3, Supplementary Materials: Video S2).

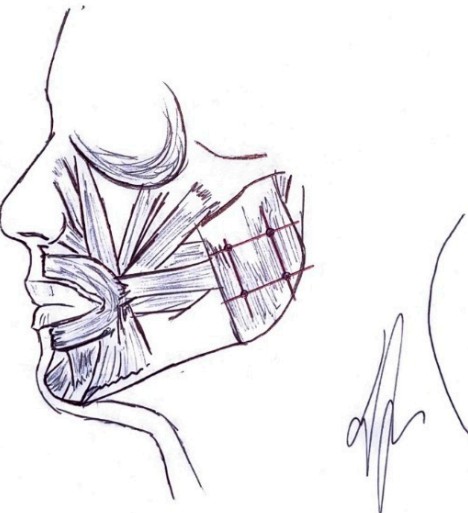

**Figure 2.** Injection site zone. A grid of 1-cm intervals is drawn on the area overlying the low and lateral side of the jaw and 2-cm from the zygomatic bone. Injections were performed on the crossed points. Four site injections per side were given.

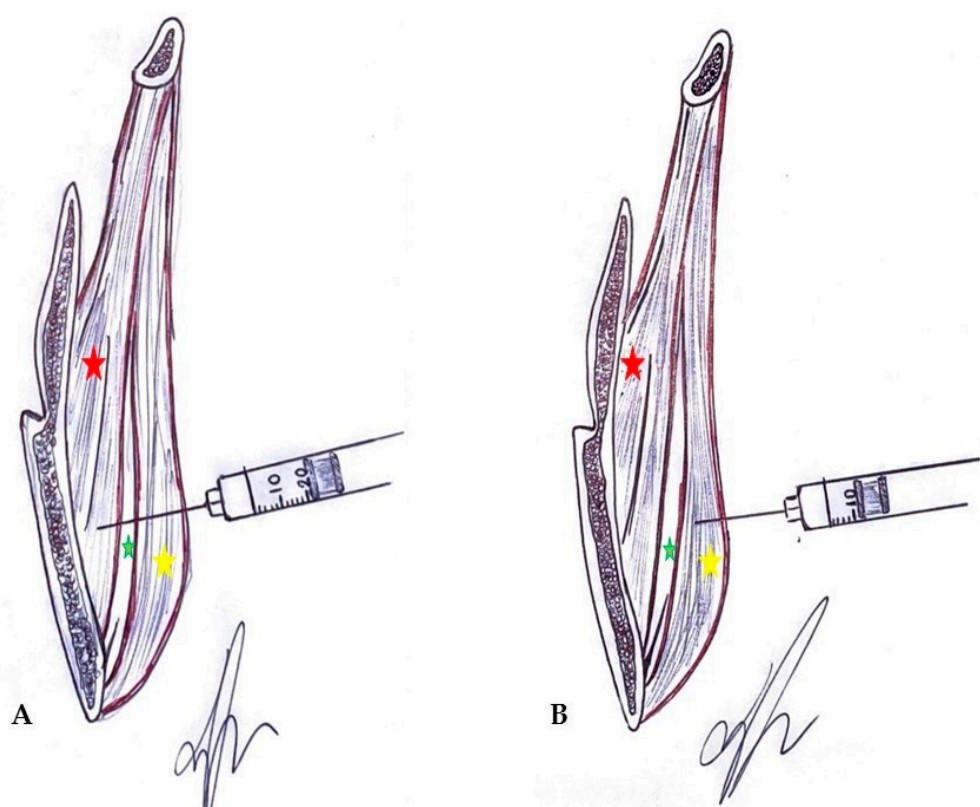

**Figure 3.** Biphasic injection technique. (**A**) 0.1 mL were released into the deep belly of the masseter muscle (red star). (**B**) 0.1 mL were released on the superficial belly (yellow star) of the masseter muscle at the same site injection. The deep inferior tendon (green star) divides the masseter muscle into two layers.

A total of 98 treatments, performed in 42 patients (23 men and 19 women; mean age, 40.5 ± 19.5 year), from 2015 to December 2020, were performed with this technique (Figure 4).

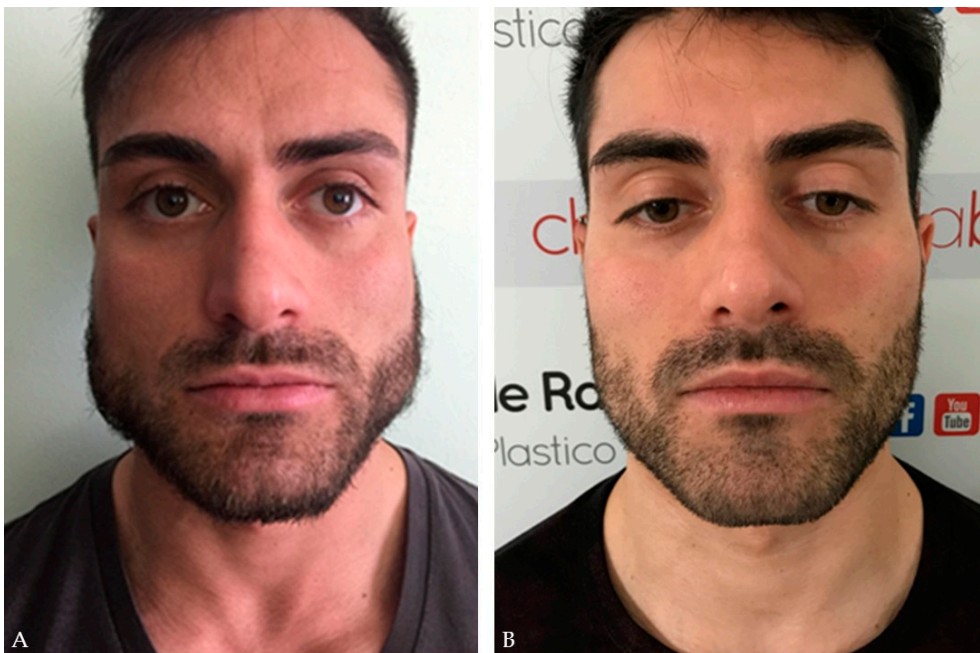

**Figure 4.** (**A**) A patient before BoNTA injection. (**B**) Follow-up of the patients after 6 months from the injection.

Patients were included in the study if they presented a widened lower facial profile with hypertrophic masseteric muscles searching for a slimmer, more aesthetic lower face. Exclusion criteria for this study included pregnancy or pregnancy attempt, breastfeeding, neuromuscular disorders (such as myasthenia gravis or Eaton-Lambert disease), peripheral neuropathy as in case of diabetes mellitus or alcoholism, concurrent administration of calcium-channel blocker, quinine, anticoagulant drugs, penicillamine, or aminoglycoside antibiotics. Written consent was obtained from all the enrolled patients. All data considered for investigation in this study were captured during pre-treatment investigations, treatment course, and post-operative visits.

Visual analog scale (VAS) was used to assess patient satisfaction: 100 represented the best possible aesthetic outcome and 0 was the worst.

In order to evaluate the BoNTA injection effect, we took periodic pictures of the patients and measured the masseter muscle reduction according to morphometric lines. The camera settings used were standardized to ensure the reproducibility of the photographs, patient after patient. The digital camera (Nikon D700, Nikon Corporation, Tokyo, Japan) fixed on a tripod, distancing 1.5 m from the patient, mounted 80 mm AF lens set on spot focus. The subject was enlightened by a 4500 Kelvin light bulb per side. The camera was set on Manual, f/8, 800 ISO scale. The patient stood up in front of the wall looking in front of him or her and the focus spot was on decided on the tragus. To calculate the masseter volumetric reduction inducted by the BoNTA injection, we took as reference a line parallel to the Frankfurt plane intersecting the lateral cantus and perpendicular to the sagittal plane passing through the middle of the nose. We drew two lines (one per each side) perpendicular to the intercantal line previously described and tangent to the zygomatic arch. According to the anatomical base of masseteric insertion to the zygomatic arch, these two lines were our reference to evaluate the masseteric morphology. A second line was drawn tangent to the masseter surface, starting from the intersection between the horizontal intercantal line and the zygomatic arch line. The angle between the zygomatic arch line and the masseter line was calculated in pre-operative and 6 months post-operative picture

to assess the reduction of masseteric projection in order to achieve lower face recontouring. Whenever the masseter line was lateral to the zygomatic arch line, we measured a positive angle, vice versa a negative angle if the first line was medial to the referring line (Figure 5).

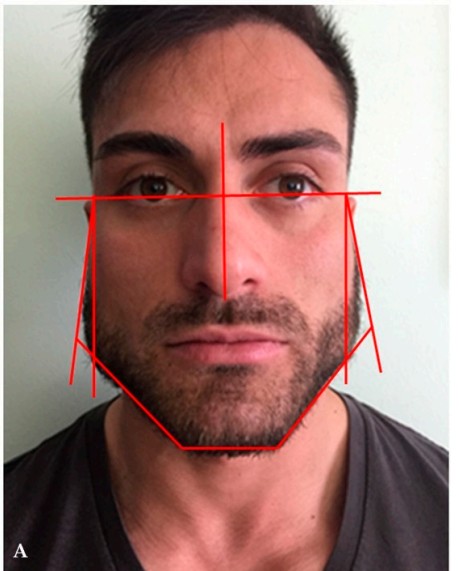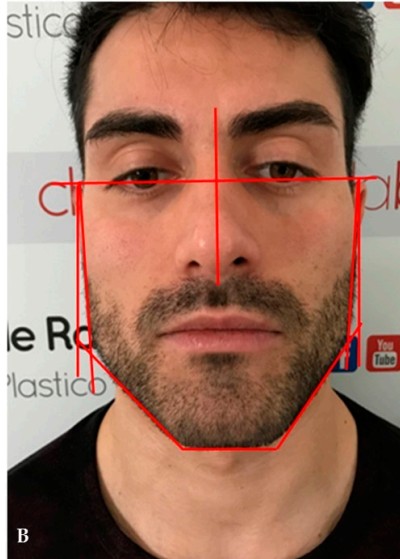

**Figure 5.** The angle between the zygomatic arch line and the masseter line was calculated in pre-operative picture (**A**) and 6 months post-operative picture (**B**) to assess the reduction of masseteric projection in order to achieve lower face recontouring.

## 3. Results

One patient experienced asymmetric smiling, which self-resolved within 4 weeks from the injection; at 1 month follow up no asymmetry was recorded. In another case, the patient claimed sagging. All the patients, if asked, claimed a mild temporary mastication force decrease for about 3 weeks after BoNTA injection, but no one claimed it as distressing. No cases of paradoxical bulging were observed (Table 1).

**Table 1.** Incidence and percentage of complications related to BonTA injection according to biphasic technique performed on patients presenting MMH.

|  | Asimmetric Smiling | Sagging | Mastication Force Decrease | Paradoxical Bulging |
|---|---|---|---|---|
| Number of Patients | 1 (2.4%) | 1 (2.4%) | 42 (100%) | 0 (0%) |

Out of 98 treatments accomplished on 42 patients, we observed a mean reduction of the left masseteric angle of $8.23° \pm 2.21°$ and a mean reduction of the right masseteric angle of $8.52° \pm 2.92°$ (Table 2). Statistical Analysis was performed using SPSS Software version 24.0 (IBM Corp., Armonk, NY, USA).

**Table 2.** Summary of the pre and post treatment angles and the overall variation (Δ), per side. Unit of measure: grade (°). Positive angles are formed by masseteric lines lateral to the zygomatic arch line. Negative angles are when the masseteric line falls medial to the perpendicular zigomatic arch line.

| Patient | LEFT SIDE PRE-OP (°) | LEFT SIDE POST-OP (°) | LEFT SIDE Δ (°) | RIGHT SIDE PRE-OP (°) | RIGHT SIDE POST-OP (°) | RIGHT SIDE Δ (°) |
|---|---|---|---|---|---|---|
| #1 | 8 | −1 | 9 | 7 | −2 | 9 |
| #2 | 4 | −1 | 5 | 3 | 0 | 3 |

**Table 2.** *Cont.*

| Patient | LEFT SIDE PRE-OP (°) | LEFT SIDE POST-OP (°) | LEFT SIDE Δ (°) | RIGHT SIDE PRE-OP (°) | RIGHT SIDE POST-OP (°) | RIGHT SIDE Δ (°) |
|---|---|---|---|---|---|---|
| #3 | 7 | −3 | 10 | 6 | −3 | 9 |
| #4 | 4 | −4 | 8 | 3 | −4 | 7 |
| #5 | 5 | −5 | 10 | 8 | −1 | 9 |
| #6 | 6 | −4 | 10 | 3 | −5 | 8 |
| #7 | 4 | −4 | 8 | 6 | −3 | 9 |
| #8 | 8 | −5 | 13 | 3 | −2 | 5 |
| #9 | 4 | −5 | 9 | 6 | −2 | 8 |
| #10 | 7 | −4 | 11 | 7 | −1 | 8 |
| #11 | 7 | −5 | 12 | 5 | −5 | 10 |
| #12 | 8 | −3 | 11 | 4 | −2 | 6 |
| #13 | 5 | −1 | 6 | 7 | −2 | 9 |
| #14 | 4 | −1 | 5 | 4 | −2 | 6 |
| #15 | 6 | −2 | 8 | 7 | 0 | 7 |
| #16 | 6 | −1 | 7 | 4 | −4 | 8 |
| #17 | 8 | −3 | 11 | 8 | −4 | 12 |
| #18 | 4 | 0 | 4 | 5 | 0 | 5 |
| #19 | 7 | −1 | 8 | 3 | −3 | 6 |
| #20 | 6 | −2 | 8 | 8 | −4 | 12 |
| #21 | 3 | 0 | 3 | 7 | −3 | 10 |
| #22 | 3 | −3 | 6 | 3 | 0 | 3 |
| #23 | 7 | −2 | 9 | 5 | −3 | 8 |
| #24 | 6 | −3 | 9 | 8 | −5 | 13 |
| #25 | 11 | −1 | 12 | 6 | −2 | 8 |
| #26 | 7 | −4 | 11 | 11 | −4 | 15 |
| #27 | 8 | −1 | 9 | 6 | 0 | 6 |
| #28 | 6 | −5 | 11 | 10 | −1 | 11 |
| #29 | 5 | −5 | 10 | 4 | 0 | 4 |
| #30 | 6 | −1 | 7 | 10 | −3 | 13 |
| #31 | 5 | −5 | 10 | 9 | −4 | 13 |
| #32 | 8 | −5 | 13 | 7 | −3 | 10 |
| #33 | 4 | −2 | 6 | 9 | −4 | 13 |
| #34 | 8 | −1 | 9 | 4 | 0 | 4 |
| #35 | 8 | −1 | 9 | 6 | −3 | 9 |
| #36 | 6 | −5 | 11 | 7 | −2 | 9 |
| #37 | 9 | −1 | 10 | 7 | −4 | 11 |
| #38 | 6 | −1 | 7 | 5 | −4 | 9 |
| #39 | 4 | −2 | 6 | 5 | −2 | 7 |
| #40 | 8 | −1 | 9 | 8 | −4 | 12 |
| #41 | 6 | −1 | 7 | 10 | −4 | 14 |
| #42 | 9 | −4 | 13 | 9 | −1 | 10 |

VAS evaluation showed no patient indicated a score of <75. Thirty-two patients gave scores of 100, six patients rated the effect from 90–99, and two patients indicated a score of 80–89. The remaining two patients gave a score of 75–79 (Table 3).

**Table 3.** Patient satisfaction. Visual analog scale (VAS) was used to assess patient satisfaction: 100 represented the best possible aesthetic outcome and 0 was the worst.

| VAS Score | 0–74 | 75–79 | 80–89 | 90–99 | 100 |
|---|---|---|---|---|---|
| Number of Patients | 0 | 2 | 2 | 6 | 32 |

## 4. Discussion

Botulinum toxin type A injection for masseter muscle reduction is a widely used procedure with an optimal therapeutic effect and favorable safety feature. Our biphasic injection approach further decreases the incidence rate of complications described in the literature [46,50], according to the unremarkable complications reported in our patients. Injections are performed in a grid-like fashion describing a safety zone that preserve the most crucial locoregional structures. Even if neurapraxia and xerostomia are reported as complications [57], in our study, no cases occurred. Moreover, no cases of loss of full

smile [50] or difficulty in mouth opening were described in our experience. Even if the most reported side effect is a decreased chewing power [8–50], in this study, no patient claimed transient muscle weakness as distressing. Moreover, none of our patient experienced bruising, the second most frequent complication reported in the literature [50]. Differently from what was reported by Xie et al., no cases of dizziness and headache occurred in our practice [58]. A case of asymmetric smiling self-resolved within a week, without the need for further BoNTA injection as described by Peng and Peng [50]. Only one patient claimed sagging; this finding was consistent with previously described studies but with a lower incidence [57,58]. Paradoxical bulging is described by several authors as a reported complication related to injection level [55,56]. However, no cases of paradoxical bulging were reported with our biphasic injection approach. The technique is tailored according to masseteric muscle anatomy injecting both deep and superficial bellies, thus avoiding the risk of muscular fiber overactivity.

Regarding the operative method, several approaches are described with important differences determined by the number of sites of injection and the injective technique. In the technique described by Kim et al., a reference line is drawn from tragus to the mouth and BoNTA is administrated into 2 points, over the masseteric prominence. Two additional points are injected, 1 cm above and 1 cm below the reference line [59]. As reported in the literature [58,60], we consider this approach risky for a sunken cheek appearance especially in patients with high middle third projection, because 2 points of injection are just below zygomatic arch. Otherwise, a two-point technique is described with injection points placed in the center of masseteric area [41]. Moreover, variation of this technique includes injecting 3 to 4 sites, 1.5 cm above the angle of the mandible [8]. Yu et al. use a distributed injection points technique: six sites are marked in a grid-like fashion [57]. A tree-point technique is described by Cheng: the first injection is delivered into the thickest portion of the muscle and subsequently the other two points are placed anteriorly and posteriorly to the first injection in a triangular configuration [61]. Choe evaluated masseteric thickness following an injection at a single site [62]. This study demonstrated that the effect of this injective pattern is extensively different between the injection site and the rest of the masseteric muscle. This evidence could consequently result in a significant difference in the volume of muscular thickness decrease thus could lead to an asymmetric lower face reshaping. In fact, in our experience, injecting evenly at 4 multiple sites has demonstrated to be much more effective, ensuring more natural results (Figure 4). Thus, we prefer the 4-point biphasic injection technique. A safe area for the injections was established marking of the injection site, a grid of 1-cm intervals is drawn on the skin overlying the low and lateral side of the jaw and a grid of 2-cm up from the zygomatic arch. This is considered a safety zone because there are no important anatomical structures. Noteworthy, staying at 1 cm far from the anterior and inferior border of the masseter ensures that we could avoid the undesired effect of paralyzing risorius and zygomatic muscles and the risk of marginalis mandibulae nerve involvement. According to our technique, care is taken to do not inject just below zygomatic bone, which can lead to a sunken cheek appearance [57]. Injecting four points versus two or one site can be paramount to avoid an unnatural facial appearing. In fact, asymmetric smile may develop because of the BoNTA spreading to adjacent surrounding muscles. Therefore, massaging masseteric muscle immediately after injection is not advisable. Further, as described by Gart [28] and according to our biphasic technique, injecting deep into the muscle bellies will reduce the incidence of undesirable toxin diffusion. Several authors suggest delivering BoNTA with the needle inserted to its full depth, asserting that superficial injections may cause asymmetry during animation [63]. Our surgically based biphasic technique, according to our cranio maxillo-facial surgery experience and anatomic knowledge, let us to affirm and suggest injecting both deeply and superficially, in order to avoid masseteric protrusion in dynamic.

According to aesthetic facial recontouring, the beauty standard tends to be globalized. Nevertheless, Asian population have unique facial features and different aesthetic cultures and compared with Westerns. The proposed lower face recontouring concept is planned

for Italian population that differs from Asian facial features. Asians have a wide, squared face compared to our study group that typically have a more angled jawline. This evidence is consistent with the evidence that a well-developed masseteric muscle is a condition more frequent in Asians than in Westerns. Furthermore, the distance between the masseters of Westerns is 105–109 mm, whereas the average for Asians is 118–125 mm, showing that Asians have an approximately 12–20 mm wider lower face than Westerns [8,9]. In some Asian countries, certain facial features are believed to be lucky. Moreover, the social pressure for a smaller and slimmer facial appearance along with cultural preference have increased the interest in lower face reshaping in many Asian nations [8] but some considerations should be underlined. The Asian face should preserve the original facial identity during BoNTA recontouring. The lower face should be treated with botulinum toxin to achieve a harmonious facial profile according to respect ethnic uniqueness. The approach to lower facial rejuvenation should be aimed according to racial characteristic of hypertrophic masseter and squared, flat jaw region. Asian patients presented a wide lower face prefer to look like attractive Asians, rather than good-looking Caucasians. In fact, sunken cheeks appearance after BoNTA injection was considered by our patient as a sign of thinness or physical wellness. On the contrary, in Asians it is usually considered as a sign of fatigue or malnourishment and thus interpreted as complication. Therefore, in those patients the superior and anterior area of the masseter should be avoided when performing BoNTA injection, as the atrophy of this region will cause a sunken cheek shape. Comparing our results with findings in Asian ethnicities, injection over the lower part of the masseter is recommended to prevent sunken lateral cheeks. An upper margin far from zygomatic arch should be set for an injection safety zone. In fact, patients should be carefully selected to achieve optimal cosmetic outcome with BoNTA injection procedure. Therefore, patients with large zygoma should be counselled appropriately before injection that can sometimes make the middle third appear more pronounced in relation to the atrophied masseters. In this situation, adjunctive remodeling options, such as lipoinjection or zygomatic bone recontouring is described [61]. This situation is reported in Asian population rather than our Italian study group where did not occur this evidence. Thus, no further remodeling approaches were required after BoNTA injection. Regardless of sex, a female aspect is reputed to be nice-looking in facial features, thus also males seek aesthetic lower facial recontouring [64]. Furthermore, male ideals of attractiveness are more in agreement with a squared jawline [64,65]. Moreover, women find this feature more masculine and attractive, even if authors contradict the dichotomous standards of beauty between the sexes [66]. Although in the literature it is reported a much lower rate of the male patients seeking for procedure (7%) [8], in our experience males requested lower face reshaping slightly more than females.

Nowadays, BoNTA masseteric hypertrophy treatment is a reliable technique because it has been performed for a long time and several protocols for an elective and safe procedure have been proposed [38–45]. Nevertheless, in the literature, several reports still describe how patients continue to experience troublesome and bothersome side effects [44–46,50].

Even if BoNTA injection in lower face recontouring is considered to be a safe and effective technique, the risk of side effect still remains [51,67], thus surgeons performing this procedure should be able to identify different complications and especially their prevention. According to our approach, no serious or worrisome side effects was reported to be caused by the treatment. As described by several studies, the most common side effect observed was indeed the temporary decrease in chewing power [8–50]. In all our patients, a mild temporary mastication force was claimed even if no one reported discomfort in mastication. It resulted from the weakened contraction of masseter muscle as an effect of decreased masseteric strength. Decreased chewing power is reported to occur 2–4 weeks after the injection, peaks at 3 weeks, and may persist for 1–2 months [8–57]. This side effect is expected because this approach is based on the toxin effect to paralyze the muscle, causing degenerating atrophy of the masseter. In our experience, it continued for 3 weeks, differently from what described in the literature. Reported complications with a

nonmuscular etiology including bruises and hematomas did not occur in our experience. Self-limited bruising was the second most common side effect described in the literature at a rate of 2.5% [50], even if it was not experienced in none of our cases. Moreover, dizziness and headache are unusual side effects. As described by Cheng [61], patients who report headaches are likely to experience headaches again after future injections, indicating an individual susceptibility. The literature describes a remarkable pattern of dose-related headache; thus, it is suggested to avoid injecting deeply in multiple sites in patients who are prone to headaches [2,50]. Differently from the reported data, in our 4-point injective biphasic technique, no cases of headache were reported. We suggest the injections should be performed from a zone 2 cm beneath the zygomatic arch to the inferior mandibular border. We believe that injecting right beneath the zygomatic bone may cause a change in facial expressions because it directly affects the mimic muscles attached to the bones, as reported by Lindern et al. [1]. Asymmetric facial expression is an undesired complication and could be due to weakened zygomatic major, zygomatic minor, and risorius, which act in lifting the corners of the mouth. This complication may occur when the point of injection is located on areas that could directly affect the major and minor zygomaticus muscles. This can easily be prevented by a proper injection technique, avoiding injection on the zygomatic bone and near the anterior border of the masseter. In fact, according to our injective pattern, we experienced asymmetric smiling in a percentage of only 2.4%, self-resolved within 4 weeks from the injection, and at 1 month follow up, no asymmetry was recorded. The loss of the full smile is due to toxin diffusion that could led to risorius muscle paralysis. In a cadaver study, this muscle attaches to the anterior and middle part of the masseter muscle in 95% of cases [51]. In our experience, no patient reported the loss of full smile; thus, we suggest that to prevent this complication, the physician should be familiar with muscular anatomy, set an injection safe zone at least one centimeter from the anterior border of the masseter, and keep both a superficial but also deep injection. As described in the literature, worsened jowls or sagging [58,60] could occur in middle-aged or elderly patients with prominent zygoma. As reported in our experience, the inability of the overlying skin to tight and contract fast enough to match the degree of atrophy of the muscle resulted in one case of local sagging. Difficulty in mouth opening is a rare complication caused by a paralysis of the lateral pterygoid muscle. As the masseter originates just below the zygomatic bone, in this location the lateral pterygoid muscles are deeper than masseter muscle. Thus, when performing biphasic injective technique, we must be aware that injecting too highly and deeply could cause an unwanted paralysis of the lateral pterygoid muscles. Xerostomia is a rare complication caused by toxin diffusion into parotid gland, resulting in reducing or thickening salivary secretion [57]. Parotid gland covers the posterior part of the masseter muscle, so surgeons should be aware not to inject too deep during the biphasic injection, keeping 1 cm far from the posterior margin of the masseter. Neurapraxia is an extremely rare complication caused by marginal mandibular nerve paralysis [57]. There were no cases of neurapraxia in the authors' experience with masseter toxin injections. In fact, cadaver studies describe that the marginal mandibular nerve runs about 0.1–1.0 cm from the mandibular border [54]. This area is not affected as the surgeons mark the injective safe zone correctly, even in the deep injection of the biphasic technique.

Uneven paradoxical bulging of masseteric muscle during mastication is a common but easily overlooked side effect that occurs because the toxin results in a different effect on the different belly of the muscle, according to the injective technique. Some parts of the masseteric belly may remain contracted while other parts could be affected by a decrease in contraction level. In the zone where the movement remains overactive, the movement appears to be much larger if compared with the region where the movement of the muscle has decreased. Before the introduction of the biphasic technique, in our experience this occurred in percentage of 2.4%, compared to the 5–10% described by Kim [8]. We confirm to the evidence already published in the literature assessing that this could more frequently occur when masseteric volume is thick, the skin is thin but especially when the technique

and the point of injection is not appropriate. After the introduction of our biphasic technique, no masseteric bulging was described. To prevent overcompensation complications, a combination of superficial and deep injections is recommended, particularly in patients with a history of paradoxical bulging after receiving masseter toxin injections.

Thus, the most common functional complications after BoNTA lower face reshaping are compensatory hypertrophy or masseteric bulging and many patients experience compensatory and stress hypertrophy symptoms [50]. Botulinum toxin masseter injection is an effective technique, commonly performed with pleasant results and good safety profiles, nonetheless an understanding of the muscular anatomy is vital to minimize undesirable complications. In fact, complications could result from uneven injections into a large area or a single layer of the muscle. Several surrounding structures should be considered: zygomaticus major and minor muscles might not to be considered as significant during BoNTA treatment, if the surgeon performs the procedure based on an accurate anatomical knowledge of the masseteric location and depth. Moreover, even if it is described that risorius muscle originates from the surface of the masseter, several studies have shown that this muscle is limited to within an anterior area that do not need to be injected [51]. We suggest the lower part of the masseter as optimal for BoNTA treatments because it constitutes the largest part of masseteric prominence, preventing the targeting of the parotid gland and thereby avoiding the risk of xerostomia. Moreover, we exclude the cheilion-tragus line due to the risk of parotid duct lesion [44–51,54]. Thus, a wide anatomical knowledge, appropriate dosing, injection location, and depth are all key factors in achieving the desired result with minimal complications.

Dose and injection site-related complication as asymmetry, sagging, or sunken lateral cheeks could be avoided according to an appropriate marking of the injection site as described by Rauso et al. [44,45]. Nevertheless, it is still poorly described in the literature how to prevent level-related complication as paradoxical bulging of the muscle [55,56].

Muscular bulging during mastication, also known as paradoxical bulging, is a rare complication that can arise after BoNTA injection in order to get a slimmer face and could occur within a week after injection. This clinical condition is related to a superficial overcompensation of the muscle as physio-pathological reaction to the neurotoxic weakening of the deep layer fibers [55,56]. If it is only the deep layer to be injected, it is expected that the superficial layer increases in strength, possibly resulting in hypertrophy and bulging, a bothersome condition for a patient that is looking for a procedure without any recovery time.

Based on this clinical evidence, several anatomical studies propose the mechanism underlying paradoxical masseteric bulging after BoNTA injection. In 2012, Cioffi et al. reported that there are more aponeuroses in the deeper part of the masseter [68]. In 2017, Lee et al. investigated the deep inferior tendon (DIT) originated deep to the aponeurosis of the superficial layer of the masseter muscle. This cadaveric study showed the presence of this tendinous structure located in the deeper part of the superficial masseter layer, which may block toxin diffusion from the deep layer to the superficial layer, making overcompensation more common [47]. Based on these observations, it was hypothesized that a broad tendon structure located within the superficial belly could act as a window that confines the injected toxin within the deeper muscle belly and avoid BoNTA spreading into the entire superficial layer. When masseter contracts, this could eventually result in a bulging of a part of superficial muscle belly that is unaffected by BoNTA due to the presence of the DIT. Subsequent studies confirm that the DIT has been identified as contributing to paradoxical masseteric bulging [69].

Our experience focuses on the findings of these evidence and highlights the importance of having accurate anatomical knowledge to ensure procedural safety.

The specific mechanism by which toxin relieves tendon structure has not yet been clearly identified, although previous studies based on pharmacological evidence have shown that muscle and tendon structures should be considered with the aim to prevent complications [70,71]. Paradoxical bulging may disappear within a week without any med-

ical treatment; thus, it is suggested to wait for toxin diffusion before a touch-up treatment. When paradoxical bulging persists or even worsens after 1–2 weeks, we recommend an additional BoNTA injection over the superficial layer. The deep layer of the masseter is almost vertical in direction and contraction, while the superficial layer originates more medially on the zygomatic bone, creating an oblique direction of contraction; paradoxical bulging usually takes this form. This clinical evidence supports the blocking effect of the tendinous structure between deep and superficial masseter layers [55,68].

The biphasic injection has been developed following this anatomical concept.

Ultimately, the superficial part of the masseter consists of not only the muscle belly but also the deep tendon structure: surgeons should be aware of this tendon prior to treating masseteric hypertrophy and improving lower face contouring using BoNTA injections. Knowledge of this critical anatomical feature will help to prevent paradoxical masseteric bulging. According to our experience, a biphasic technique of superficial and deep injections is recommended. Considering the biphasic injective procedure applied for masseteric hypertrophy BoNTA treatment, we could predict and avoid the possibility of compensatory hypertrophy of superficial masseter layer.

## 5. Conclusions

In conclusion, the most important consideration of this study is that an injective protocol of BoNTA should be performed in consideration of the tendon structure to avoid unexpected results.

To minimize undesirable and unwanted complications, a deep knowledge of the facial anatomy is paramount. Appropriate injection points and depth are key factors in achieving the desired and expected results with minimal complications. According to our experience, an accurate marking of the injection area is vital. It is important to locate the posterior, anterior, inferior and superior borders of the safe zone of injection, an area of 1-cm intervals drawn on the low and lateral side of the muscle, 2 cm from the zygomatic arch. Keeping injections inside the safe zone in 4 different points and performing a multilayer biphasic injective technique, is crucial for the prevention of common side effects reported in the literature, such as paradoxical bulging.

In fact, no significant side effects were noted following botulinum toxin type A injection. This may be attributable to our technique based on the results obtained in anatomical studies; thus, we recommend performing a biphasic injection into the superficial and deep muscle bellies of the masseter.

We feel that sharing this tip could be helpful for all the physicians involved in MMH treatment with BoNTA.

**Supplementary Materials:** The following are available online at https://www.mdpi.com/article/10.3390/app11146478/s1, Video S1: Muscular bulging during mastication. Paradoxical masseter bulging 1 week apart from BoNTA injections, Video S2: Biphasic Injection technique. Injections were performed with a 27 G, 4 cm length needle; 0.1 mL were released into the deep belly and 0.1 mL on the superficial belly of the masseter muscle at the same site injection. Four site injections per side were performed.

**Author Contributions:** Conceptualization, G.M.R.; methodology, F.C.; validation, R.F., G.L.G. and P.B.; formal analysis, N.D.F.; investigation, A.C.; resources, G.A.; data curation, F.C.; writing—original draft preparation, F.C.; writing—review and editing, G.M.R.; visualization, P.B., R.F., G.L.G. and A.C.; supervision, N.D.F., A.C. and G.M.R.; project administration, G.A. and F.C. All authors have read and agreed to the published version of the manuscript.

**Funding:** This research received no external funding.

**Institutional Review Board Statement:** Every patient signed an informed consent for the procedures, the use and publication of images and clinical data for scientific research purposes. Data privacy was handled according GDPR.

**Informed Consent Statement:** Informed consent was obtained from all subjects involved in the study. Written informed consent has been obtained from the patients to publish this paper.

**Data Availability Statement:** Not applicable.

**Conflicts of Interest:** The authors declare no conflict of interest.

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
