# Peer review of "Biphasic Injection for Masseter Muscle Reduction with Botulinum Toxin"

_applsci, doi:10.3390/app11146478_

Round 1

Reviewer 1 Report

The article Biphasic Injection for Masseter Muscle Reduction with Botulinum Toxin brings an interesting point of view regarding the treatment of the masseter muscle hypertrophy. I appreciate the clinical experience of the team in this topic.

However, the article must described the other techniques that are presented in the scientific literature, the advantages and the disadvantages and to compare with the proposed technique. 

The biphasic technique is very poor described in the materials and methods section. There are no inclusion or exclusion criteria of the subjects involved in the study.

The results reported are speculative and subjective, without any objective scale that can measure them.

The article doesn't have the Institutional Review Board Statement.

Author Response

Dear Reviewers

We would thank you for your feedback on our manuscript.

The points presented in your letter have been copied below, and the responses are indicated beneath each item (in blue type).

The newest content revisions to the manuscript text appear in red type in the files.

Review Report 1
=====================

English language and style

( ) Extensive editing of English language and style required
( ) Moderate English changes required
(x) English language and style are fine/minor spell check required

( ) I don't feel qualified to judge about the English language and style

                                                  Yes       Can be improved    Must be improved      Not Applicable                   

Does the introduction                   ()                 (x )                           ( )                            ( )

provide sufficient

background and include

 all relevant references?

Is the research                               ()                 ( x)                          ( )                            ( )

 design appropriate?

Are the methods                            ()                  ( )                            ( x)                         ( )

adequately described?

Are the results                                ( )                           ( )                              ( x)                        ( )

 clearly presented? 

Are the conclusions                        ( )               ()                              (x )                        ( )

supported by the results?

Comments and Suggestions for Authors

The article Biphasic Injection for Masseter Muscle Reduction with Botulinum Toxin brings an interesting point of view regarding the treatment of the masseter muscle hypertrophy. I appreciate the clinical experience of the team in this topic.

However, the article must described the other techniques that are presented in the scientific literature, the advantages and the disadvantages and to compare with the proposed technique.

The biphasic technique is very poor described in the materials and methods section. There are no inclusion or exclusion criteria of the subjects involved in the study.

The results reported are speculative and subjective, without any objective scale that can measure them.

The article doesn't have the Institutional Review Board Statement.

Response to Reviewer 1 Comments

We would thank you for your precious comments.

We have provided to describe the other techniques that are presented in literature, underling the respective drawbacks and advantages. Moreover, we have compared them with the proposed technique, as you kindly suggested.

We have provided to describe in detail the technique in the materials and methods section.

We have added inclusion and exclusion criteria of the patients involved in the study.

We have provided to add both objective and subjective evaluation of the results. However, the purpose of our study is to present the biphasic injection technique in order to prevent paradoxical bulging and not to demonstrate the efficacy of BoNTA injection in case of MMH, a topic widely described in literature.

The study design of our research is Retrospective. More specifically this retrospective study was performed in a Private Practice setting as underlined in the manuscript. Every patient signed an informed consent for the procedures. The patient also gave consent for use and publication of images and clinical data for scientific research purposes and data privacy was handled according to GDPR. All of these information is specified in the informed consent attached.

I hope that these revisions meet with your approval.

Again, we are grateful for the feedback. We look forward to your comments on the revised version.

Kind Regards,                              

Fabrizio Chirico MD

Reviewer 2 Report

The manuscript is very well written and concise, please find some minor points below:

Abstract:

-------------

Change the sentence in Line 19: "moreover, non-surgical facial slimming is often performed nowadays with multiple procedures such as 19 BoNTA injection into masseter muscle"

Line 27: "by injecting..."

Introduction:

---------------

Please indicate if there are gender or ethnicity differences in facial shape preferences?

Line 47: please indicate if MHH is a long-recognized disease?

Line 68: Please indicate when treatment with BoNT/A was first introduced?

Line 75: Therapeutic indications of BoNT/A are much wider, please list all the major indications: movement disorders, hemifacial spasm, essential tremor, tics, writer’s cramp, cervical dystonia, cerebral palsy and vascular cerebral stroke More recently, BoNT has been approved for chronic pain, migraine headache, overactive bladder, inflammation  some forms of depression...

Please include one review by Dressler D. or Jankovic. J.

Materials and Method:

------------------------------

Please include a picture showing the possible injection points, this could be helpful for practitionners.

Line 113: "deliver" instead of "delivery"

Discussion:

---------------

Line 130: "acts selectively on..." instead of "produces action..."

Line 134: "at" high concentrations

Please indicate if any side effects were observed far from the injection site and compare with other studies done with this technique.

Re-phrase of Lines 150-152: "Because of BoNTA action on nerve endings, an accurate and extensive anatomical knowledge of nerve endings of the targeted muscle is paramount to ensure maximum outcome with the minimum toxin concentration."

Lines 175-178: Too long sentence, please do 2 sentences to clarify.

Line 200: please provide more details about the side effects: is the risk similar for all patients or is there an influence of the patient physical characteristics?

In the Conclusions, please provide a detailed injection protocol describing the precise injection sites to be use in order to avoid side effects. This may help new practitionners in the field.

Author Response

Dear Reviewers

We would thank you for your feedback on our manuscript.

The points presented in your letter have been copied below, and the responses are indicated beneath each item (in blue type).

The newest content revisions to the manuscript text appear in red type in the files.

Review Report 2
=====================

English language and style

( ) Extensive editing of English language and style required
( ) Moderate English changes required
(x) English language and style are fine/minor spell check required

( ) I don't feel qualified to judge about the English language and style

                                                  Yes       Can be improved    Must be improved      Not Applicable                   

Does the introduction                   ()                 (x )                           ( )                            ( )

provide sufficient

background and include

 all relevant references?

Is the research                               (x)                ( )                            ( )                            ( )

 design appropriate?

Are the methods                            ()                  ( x)                          ( )                           ( )

adequately described?

Are the results                                ( x)                         ( )                              ( )                          ( )

 clearly presented? 

Are the conclusions                        (x )             ()                              ( )                          ( )

supported by the results?

Comments and Suggestions for Authors

The manuscript is very well written and concise, please find some minor points below:

Abstract:

-------------

Change the sentence in Line 19: "moreover, non-surgical facial slimming is often performed nowadays with multiple procedures such as 19 BoNTA injection into masseter muscle"

Line 27: "by injecting..."

Introduction:

---------------

Please indicate if there are gender or ethnicity differences in facial shape preferences?

Line 47: please indicate if MHH is a long-recognized disease?

Line 68: Please indicate when treatment with BoNT/A was first introduced?

Line 75: Therapeutic indications of BoNT/A are much wider, please list all the major indications: movement disorders, hemifacial spasm, essential tremor, tics, writer’s cramp, cervical dystonia, cerebral palsy and vascular cerebral stroke More recently, BoNT has been approved for chronic pain, migraine headache, overactive bladder, inflammation  some forms of depression...

Please include one review by Dressler D. or Jankovic. J.

Materials and Method:

------------------------------

Please include a picture showing the possible injection points, this could be helpful for practitionners.

Line 113: "deliver" instead of "delivery"

Discussion:

---------------

Line 130: "acts selectively on..." instead of "produces action..."

Line 134: "at" high concentrations

Please indicate if any side effects were observed far from the injection site and compare with other studies done with this technique.

Re-phrase of Lines 150-152: "Because of BoNTA action on nerve endings, an accurate and extensive anatomical knowledge of nerve endings of the targeted muscle is paramount to ensure maximum outcome with the minimum toxin concentration."

Lines 175-178: Too long sentence, please do 2 sentences to clarify.

Line 200: please provide more details about the side effects: is the risk similar for all patients or is there an influence of the patient physical characteristics?

In the Conclusions, please provide a detailed injection protocol describing the precise injection sites to be use in order to avoid side effects. This may help new practitionners in the field.

Response to Reviewer 2 Comments

We would thank you for your precious comments.

-Line 19-27: In abstract section, we have provided to change the sentence at line 19 and to made the suggested correction at line 27.

In introduction section, we have indicated gender or ethnicity differences in facial shape preferences, as you kindly suggested.

Line 47: we have indicated if MHH is a long-recognized disease

Line 68: we have indicated when treatment with BoNTA was first introduced

Line 75: We agree with your suggestion to list all the major indication. We have included review by

Dressler D or Jankovic. J.

In Materials and Method section, we have included a picture showing the possible injection points and corrected "deliver" instead of "delivery" at line 113.

In discussion section, we have made the suggested corrections at line 113 and line 134.

We have indicated if any side effects were observed far from the injection site and compared with other studies done with this technique, as you suggested. We have provided to rephrase at lines 150-52.

At lines 175-178, we have clarified the sentence in two periods.

At line 200, we have provided to add more details about the side effects, also respect patient physical characteristics.

In the conclusions, we have provided a detailed injection protocol describing the precise injection sites to be use in order to avoid side effects.

Again, we are grateful for the feedback. We look forward to your comments on the revised version.

Kind Regards,

Fabrizio Chirico MD

Reviewer 3 Report

Topic is not novel, thus the manuscript does not add anything new into body of evidence in terms of ONBTX-A injections in TMD patients. Also, there are so many major flaws in the study that eventually renders the manuscript unpublishable. Hence, I have some ethical concerns whether authors should be doing this to their patients anymore as they seemingly lack the very basic textbook knowledge in this field. See below:

Abstract

L16-17 - MMH of unknown aetiology? In 2021 it reads like a sad misunderstanding - there were tons of papers published in terms of masticatory muscles and their hypertophy/hyperplasia contributiong factors

Also abstract structure reads like a mini introduction section in humanities - this section should be structurized according to general MDPI scientific abstract guidelines

Introduction

L47 - chronic bruxisms?

L73 - there is no such thing as 'bruxism treatment' - 'management' is the word Authors are missing

L108-109 - these results were published already and having just 21 patients from 2006 to 2014, with a total of 43 treatments performed on 21 patients by a senior author - firstly there is either not sufficient number of patients to provide sound scientific evidence, yet they were not compared. Later, authors claim that 'total of 98 treatments, performed in 42 patients, from 2015 to December 2020 have been performed with this technique'. I also do not happen to understand the contribution of 7 additional authors.

L115-116 - these brand names are missing manufacturers' data

L117 - method needs detailed description, as the needle type requires manufacturers' data

Results

This section is missing everything in terms of scientific evidence - there are no numbers, tests, tables, graphs, charts, analyses provided... are authors sure they picked the right Journal for this type of publication?

Dicussion

It reads like a literature review - it needs to be re-written according to newly formulated results, with conclusions section forthcoming  - according to the manuscript Authors provided no scientific merit can be drawn

Author Response

Dear Reviewers

We would thank you for your feedback on our manuscript.

The points presented in your letter have been copied below, and the responses are indicated beneath each item (in blue type).

The newest content revisions to the manuscript text appear in red type in the files.

Review Report 3
=====================

English language and style

( ) Extensive editing of English language and style required
( ) Moderate English changes required
(x) English language and style are fine/minor spell check required

( ) I don't feel qualified to judge about the English language and style

                                                  Yes       Can be improved    Must be improved      Not Applicable                   

Does the introduction                   ()                 ( )                             ( x)                          ( )

provide sufficient

background and include

 all relevant references?

Is the research                               ()                 ( )                            ( x)                          ( )

 design appropriate?

Are the methods                            ()                  ( )                            (x )                         ( )

adequately described?

Are the results                                ( )                           ( )                              (x )                        ( )

 clearly presented? 

Are the conclusions                        ( )               ()                              ( x)                        ( )

supported by the results?

Comments and Suggestions for Authors

Topic is not novel, thus the manuscript does not add anything new into body of evidence in terms of ONBTX-A injections in TMD patients. Also, there are so many major flaws in the study that eventually renders the manuscript unpublishable. Hence, I have some ethical concerns whether authors should be doing this to their patients anymore as they seemingly lack the very basic textbook knowledge in this field. See below:

Abstract

L16-17 - MMH of unknown aetiology? In 2021 it reads like a sad misunderstanding - there were tons of papers published in terms of masticatory muscles and their hypertophy/hyperplasia contributiong factors

Also abstract structure reads like a mini introduction section in humanities - this section should be structurized according to general MDPI scientific abstract guidelines

Introduction

L47 - chronic bruxisms?

L73 - there is no such thing as 'bruxism treatment' - 'management' is the word Authors are missing

L108-109 - these results were published already and having just 21 patients from 2006 to 2014, with a total of 43 treatments performed on 21 patients by a senior author - firstly there is either not sufficient number of patients to provide sound scientific evidence, yet they were not compared. Later, authors claim that 'total of 98 treatments, performed in 42 patients, from 2015 to December 2020 have been performed with this technique'. I also do not happen to understand the contribution of 7 additional authors.

L115-116 - these brand names are missing manufacturers' data

L117 - method needs detailed description, as the needle type requires manufacturers' data

Results

This section is missing everything in terms of scientific evidence - there are no numbers, tests, tables, graphs, charts, analyses provided... are authors sure they picked the right Journal for this type of publication?

Dicussion

It reads like a literature review - it needs to be re-written according to newly formulated results, with conclusions section forthcoming  - according to the manuscript Authors provided no scientific merit can be drawn

Response to Reviewer 3 Comments

We would thank you for your precious comments.

We would kindly clarify that the purpose of our study is to present the biphasic injection technique in order to prevent complications as paradoxical bulging, a topic poorly described in literature. Thus, the aim is not to demonstrate the efficacy of BoNTA injection in case of MMH: this field of research is already extensively discussed by a wide amount of scientific works.

Moreover, the manuscript is not about BoNTA injections in TMD patients, but the scope is to present a technique in order to minimize undesirable complications as masseteric bulging in lower face recontouring.

Our effort in this revision is about the major flaws you kindly suggested, according to our cranio-maxillo facial surgical knowledge in this field.

We have provided to improve abstract, structured according to general MDPI scientific guidelines.

About MMH aetiology, the most recent literature still demonstrate how MMH causes are controversial. Moreover, several highly cited papers (Choe et al., Park et al., Shome et al, Rispoli et al. , etc.)  define MMH aetiology as unknow. However, we have provided to underline the contributing factors as you kindly suggested.

In introduction section, we have provided the suggested revisions at line 47 and 73. 

The contributions of the additional authors is explained in their rules in this study, more specifically: conceptualization, methodology, validation, formal analysis, investigation, resources, data curation, writing original draft and editing, supervision, and project administration.

At line 115-116-117, we added the suggested detailed descriptions of the method as manufactures’ data.

In results section, we provided to add numbers, tables and analyses of the outcome, as suggested. Discussion was re-written according to newly formulated results with conclusions section forthcoming. 

Again, we are grateful for the feedback. We look forward to your comments on the revised version.

Kind Regards,

Fabrizio Chirico MD

Round 2

Reviewer 1 Report

Dear authors,

Congratulations for the revised paper. Now, the manuscript is well written and can be published.

The only modification that must be done is on the figure 3. I think that the figure A and B are reversed. (A) A patient before BoNTA injection. (B) Follow-up of the patients after 6 months from the injection. Please verify, i think that B is the patient before the injection and A is the patient after the injection.

Author Response

Dear Reviewers

We would thank you for your feedback on our manuscript.

The points presented in your letter have been copied below, and the responses are indicated beneath each item (in blue type).

The newest content revisions to the manuscript text appear in red type in the files.

Review Report 1
=====================

English language and style

( ) Extensive editing of English language and style required
( ) Moderate English changes required
(x) English language and style are fine/minor spell check required

( ) I don't feel qualified to judge about the English language and style

                                                  Yes       Can be improved    Must be improved      Not Applicable                   

Does the introduction                   (x)               ( )                             ( )                            ( )

provide sufficient

background and include

 all relevant references?

Is the research                               (x)                ( )                            ( )                            ( )

 design appropriate?

Are the methods                            (x)                 ( )                           ( )                           ( )

adequately described?

Are the results                                (x)                 ( )                           ( )                          ( )

 clearly presented? 

Are the conclusions                        (x )             ()                              ( )                          ( )

supported by the results?

Comments and Suggestions for Authors

Dear authors,

Congratulations for the revised paper. Now, the manuscript is well written and can be published.

The only modification that must be done is on the figure 3. I think that the figure A and B are reversed. (A) A patient before BoNTA injection. (B) Follow-up of the patients after 6 months from the injection. Please verify, i think that B is the patient before the injection and A is the patient after the injection.

Response to Reviewer 1 Comments

We would thank you for your precious comments and we would apologize for the mismatch in the figure.

We have provided to correct the sequence in the picture as you kindly suggested.

I hope that these revisions meet with your approval.

Again, we are grateful for the feedback.

Kind Regards,                              

Fabrizio Chirico MD

Reviewer 3 Report

The manuscript was greatly improved, as most of my remarks were successfully conveyed. However some issues still persist:

Abstract section should be re-formulated after addressing the latter remarks. See below:

Introduction

L41-L81 - an article and a study group is described as Italian origin, hence there are too many mentions about 'Asians' vs 'Westerners' in this part - please re-formulate and try using these citations rather in Discussion section to compare your own results with findings in other ethnicities.

L86-87 - there are almost none of non-surgical treatments described in terms of bruxism and TMDs- probably due to the fact, that majority of the Authors work at Maxillofacial surgery unit. Hence I recommend incorporating and citing: https://pubmed.ncbi.nlm.nih.gov/25439631/ , https://pubmed.ncbi.nlm.nih.gov/34071832/ also those two will provide excellent rationale prior to hypothesis formulation - they are among of the most important papers published recently in terms of ONBTX-A and complex bruxism management https://pubmed.ncbi.nlm.nih.gov/24634627/ and  https://pubmed.ncbi.nlm.nih.gov/32182879/

L148 - despite all the Pictures and Figures provided Authors should include a photograph of this 'paradoxical bulging'

L164 and cont'd- please carefully check brand and factory names names for consistency with MDPI policy in this matter e.g. 'onabotulinumtoxin A (ONA; Botox, Allergan Inc., Irvine, CA, USA)' should be mentioned as follows: Botox (Allergan, an AbbVie company, North Chicago, Illinois, US); and so on

L167 - 0.9% saline

L198-204 Figure 3 A and B are mismatched. However, to keep things straight - Figure 3,4 and 5 - it is difficult for me to comment on this. Do not get me wrong - I am not an adversary of using photo editing software - in a wise way. These are obviously overedited - please provide good quality pictures 'before' and 'after' with exact timespan. Raw files preferred. Do not alter shape and sizes - keen eye will always notice and treatment effects are visible without special FX. Also make sure that pictures are taken under the same angles, background and lighting conditions. I feel that superimposition is completely redundant, too. 

L256 - did the authors mean 'biphasic'?

Discussion

Initial part of this section (L281-L303) reads like an Introduction and should be moved in there. On the contrary, I have previously highlighted parts of the Introduction section which should be moved into the Discussion - this section should start with brief summary of Authors' own findings, with comparison with other work followed.

L393 - biphasic?

L500 - 'in our experience' is redundant

Author Response

Dear Reviewers

We would thank you for your feedback on our manuscript.

The points presented in your letter have been copied below, and the responses are indicated beneath each item (in blue type).

The newest content revisions to the manuscript text appear in red type in the files.

Review Report 3
=====================

English language and style

( ) Extensive editing of English language and style required
( ) Moderate English changes required
(x) English language and style are fine/minor spell check required

( ) I don't feel qualified to judge about the English language and style

                                                  Yes       Can be improved    Must be improved      Not Applicable                   

Does the introduction                   ()                 ( )                             ( x)                          ( )

provide sufficient

background and include

 all relevant references?

Is the research                               ()                 ( x)                          ( )                            ( )

 design appropriate?

Are the methods                            (x)                   ( )                         ()                            ( )

adequately described?

Are the results                                ( )                           ( )                              (x )                        ( )

 clearly presented? 

Are the conclusions                        ( )               (x)                            ( )                          ( )

supported by the results?

Comments and Suggestions for Authors

The manuscript was greatly improved, as most of my remarks were successfully conveyed. However some issues still persist:

Abstract section should be re-formulated after addressing the latter remarks. See below:

Introduction

L41-L81 - an article and a study group is described as Italian origin, hence there are too many mentions about 'Asians' vs 'Westerners' in this part - please re-formulate and try using these citations rather in Discussion section to compare your own results with findings in other ethnicities.

L86-87 - there are almost none of non-surgical treatments described in terms of bruxism and TMDs- probably due to the fact, that majority of the Authors work at Maxillofacial surgery unit. Hence I recommend incorporating and citing: https://pubmed.ncbi.nlm.nih.gov/25439631/ , https://pubmed.ncbi.nlm.nih.gov/34071832/ also those two will provide excellent rationale prior to hypothesis formulation - they are among of the most important papers published recently in terms of ONBTX-A and complex bruxism management https://pubmed.ncbi.nlm.nih.gov/24634627/ and  https://pubmed.ncbi.nlm.nih.gov/32182879/

L148 - despite all the Pictures and Figures provided Authors should include a photograph of this 'paradoxical bulging'

L164 and cont'd- please carefully check brand and factory names names for consistency with MDPI policy in this matter e.g. 'onabotulinumtoxin A (ONA; Botox, Allergan Inc., Irvine, CA, USA)' should be mentioned as follows: Botox (Allergan, an AbbVie company, North Chicago, Illinois, US); and so on

L167 - 0.9% saline

L198-204 Figure 3 A and B are mismatched. However, to keep things straight - Figure 3,4 and 5 - it is difficult for me to comment on this. Do not get me wrong - I am not an adversary of using photo editing software - in a wise way. These are obviously overedited - please provide good quality pictures 'before' and 'after' with exact timespan. Raw files preferred. Do not alter shape and sizes - keen eye will always notice and treatment effects are visible without special FX. Also make sure that pictures are taken under the same angles, background and lighting conditions. I feel that superimposition is completely redundant, too.

L256 - did the authors mean 'biphasic'?

Discussion

Initial part of this section (L281-L303) reads like an Introduction and should be moved in there. On the contrary, I have previously highlighted parts of the Introduction section which should be moved into the Discussion - this section should start with brief summary of Authors' own findings, with comparison with other work followed.

L393 - biphasic?

L500 - 'in our experience' is redundant

Response to Reviewer 3 Comments

We would thank you for your precious comments.

We have re-formulated abstract section after addressing the remarks you kindly suggested to us.

We have re-formulate the part at line 41-81 in discussion section, comparing our results with findings in other ethnicities.

We would thank you for the precious comment at line 86-87. The studies you suggested are brilliant and really remarkable in this field. Thus we are pleased to incorporated and citing those experience in the body of the manuscript.

We have included a photograph of paradoxical masseter bulging as you kindly recommended.

At line 164 and continued, we have carefully checked brand and factory names for consistency with MDPI policy in this matter.

At line 167, we have made the suggested correction.

At line 198-204, we would apologize for the mismatch in the figure. We have provided to correct the sequence in the picture as you kindly suggested. Nevertheless, the pictures were not edited, just cropped to focus the attention on the lower third of the face. For sure, your advice to take pictures under the same light, angle and background is preferable but, even though the pictures were taken with the same setting, they were taken in-office not in a photographic studio. For this reason, the pictures may change light in response to the surrounding light that is not completely compensated by the flash lights. In the last months, we adopted the new background that you can see in Figure 4B (6 months post). For this reason, we do not have pre and post treatment pictures with the same background. Regarding the request for a RAW extension of the pictures file, we would really like to follow your suggestion but, due to its great dimensions (>40mb each), it is not compatible with the maximum size of the uploadable files and it would not influence the quality of the results, it would need indeed a post-production process to edit the colours. We thought that a superimposition of the pre and post figure would have been eye catching but, as you kindly recommended, treatment effects are visible even without it.

At line 256, we have provided the requested correction

We have provided to improve discussion section. We have added a brief summary of our findings in comparison with other work followed, as you suggested. Moreover, we have moved the part at line 281-303 in introduction section, as you recommended.

At line 393 and 500, we have provided the requested correction.

Again, we are grateful for the feedback and we are really thankful for all your recommendations. These have been precious to improve the quality of our work.

We look forward to your comments on the revised version.

Kind Regards,

Fabrizio Chirico MD
